# Recent Progress in Modulated Photothermal Radiometry

**DOI:** 10.3390/s23104935

**Published:** 2023-05-20

**Authors:** Javier Corona, Nirmala Kandadai

**Affiliations:** College of Engineering, Oregon State University, Corvallis, OR 97331, USA; coronjav@oregonstate.edu

**Keywords:** photothermal, radiometry, modulated, optic, conductivity

## Abstract

In this review, the emerging work using a technique known as modulated photothermal radiometry (MPTR) is evaluated. As MPTR has matured, the previous discussions on theory and modeling have become increasingly limited in their applicability to the current state of the art. After a brief history of the technique, the currently used thermodynamic theory is explained, highlighting the commonly applied simplifications. The validity of the simplifications is explored via modeling. Various experimental designs are compared, and the differences are explored. New applications, as well as emerging analysis techniques, are presented to emphasize the trajectory of MPTR.

## 1. Introduction

Thermal processes are fundamental to electrical, mechanical, chemical, and biological fields [1]. Quantifying the heat transfer dynamics and the resulting temperature distribution is crucial in determining the physical, electrical, and chemical characteristics of materials. Thermophysical properties, such as thermal conductivity and heat capacity, dictate the material’s thermal processes [2]. Early efforts by researchers such as Shanks et al. [3] measured the thermal dynamics directly. Shanks et al. measured the transient temperature distribution with thermocouples placed along the depths of silicon samples. Boundary heating was accomplished with electric heaters at either end. While this method is straightforward to apply, it suffers from various limitations, such as reduced accuracy due to probe heat loss, large sample size requirements (larger than a few cm), and significant measurement time (hours) [4]. These issues are exacerbated when samples have nonuniform thermal properties throughout their depths.

The 3ω technique was introduced to combat these issues [5]. In this method, a periodic current is passed through a deposited probe, resulting in a heat wave that diffuses through the sample. The voltage across the probe is measured and related to the temperature-dependent resistivity of the metal. By precise tuning of the modulation frequency, the penetration of the wave can be controlled; thus, the depth-dependent thermal properties can be measured without requiring additional thermal probes. The developed temperature at the sensor can then be related to the thermal properties of the sample [6]. The 3ω method was effective in reducing heat losses from the reduced probe size and was successful in the measurement of thermal properties in thin, single, and multilayer film [6,7]. 

Despite the success of the 3ω technique, a noncontact technique has been sought in many applications [4]. A technique known as thermoreflectance was introduced to address this need [8,9]. In this method, a thin metal is evaporatively deposited onto a sample of interest. An ultrafast (ps or less) pulsed laser source is split into two low- and higher-power beams, called probe and pump beams, respectively. The pump heats the surface, causing a time-dependent temperature change. The reflection of the probe beam by the sample surface is monitored over a long time period. The temperature–reflectance relationship can be used to precisely calculate the thermal properties of the sample. While this technique simplified sample preparation and enabled the study of ultrafast phenomena, it suffers from various challenges, such as weak thermoreflectance coefficients and complex hardware and data analysis [10]. Again, this led other researchers to explore different methods.

Modulated photothermal radiometry (MPTR) was first introduced by Nordal and Konstal [11] in 1979. In their experiment, a continuous-wave CO_2_ laser source (emission wavelength of 10.6 µm [12]), modulated by an optomechanical chopper, was used to periodically heat potassium sulfate powders. Infrared optics focused the resulting thermal emission into an infrared detector, where a lock-in amplifier and radiometer recorded the emission spectrum of the powder bed. Using the theoretical background of photoacoustic spectroscopy and the Stefan–Boltzmann law, the absorption spectrum was related to the recorded emission. While Nordal and Konstal’s work lacked a detailed discussion on the measured spectra and only weakly attributed the spectral peaks to the temperature-induced structural changes, they demonstrated the potential of MPTR for the remote spectroscopic analysis of powders, coatings, and other surface-based processes with minimal equipment and sample preparation. 

Later work sought to extend the applications of MPTR. Santos and Miranda introduced the first comprehensive MPTR theory on solids and effectively showed how the MPTR signal could compute the thermophysical properties [13]. Their contribution expanded the thermal analysis of MPTR to include atmospheric effects and developed specific analyses for thermally thin and thick samples. The field of MPTR grew as efforts were made to increase the theoretical understanding with modeling, to simplify the analysis, and to extend the application to a wider set of materials [14,15,16,17]. Nakamura et al. [18] used MPTR to study and characterize the quality of semiconductor wafers [18]. By scanning the sample surface, the MPTR signal magnitude for each position was created and used to produce images that effectively highlighted the defects in GaAs wafers. Mandelis et al. [19] showed that the quantitative characterization of semiconductors was possible by providing the theoretical framework for MPTR in solid-state electronics. They later reinforced Nakamura’s observations by providing an in-depth description of semiconductor depth profilometry to identify subsurface defects [20], showing its potential to monitor the fabrication process in the semiconductor industry. Batista et al. [21] used the developed theory to measure photocarrier properties, such as lifetime, across a wide temperature range with MPTR. 

The focus shifted away from semiconductors as other researchers moved toward using MPTR to measure microfeatures. It was observed that the depth resolution of MPTR could be increased by using materials with high optical attenuation. Battaglia et al. [22] set out to show this by using MPTR to characterize copper oxide deposits down to 750 nm in thickness. Their success reinforced the MPTR application to thin films. 

Present-day MPTR is highly sensitive to the intrinsic properties of materials. Optical properties such as absorption cross-section and skin depth determine the amount of heat absorbed and the depth of the optical penetration [23] at the incident surface. In electronic materials such as silicon and germanium, the generation of photocarriers through a photon–electron interaction can also contribute to the thermal response through nonradiative relaxation to the ground state [24]. The associated temperature increase results in surface emission. The emission intensity and spectrum are consistent with Plank’s black body radiation [25]. Radiative relaxation of generated photocarriers can also contribute to the emission, resulting in small spectral peaks in addition to the black body radiation [15]. The intensity of the light, as it passes through the material, decreases exponentially in accordance with the material’s attenuation coefficient [26]. This causes an axial and radial heat flow in the material and a time-varying temperature distribution defined by Fourier’s law of heat conduction [27]. When considering the periodic heating conditions from the modulated beam intensity, the expected temperature response is also periodic and consists of constant (DC) and varying (AC) components. The DC component is generated from the steady-state response of the material to the average power of the modulated light. The AC components result from the frequency modulation of the laser. The amplitude and phase of these AC components are dictated by the properties of the light and the thermophysical properties of the material. The periodic heating decays exponentially through the depth of the material [27]. The radiometric systems of the MPTR collect the emission and relate the surface temperature to the material properties and allow the monitoring of underlying thermal responses without any contact or disturbance of the system. The material generates a thermal signal that is dependent on the thermal conductivity, layer thickness, density, and heat capacity.

In our review, we expand on the previous review on this subject that was conducted by Pham et al. in 2014 [28]. The previous review showcased applications of MPTR that only used the phase of the thermal response to calculate the material’s thermal properties. Since then, researchers have investigated and realized a simple analysis that calculates thermal properties using only the signal’s amplitude. In our work, we describe the theory and modeling of the system that allows simplified analysis. This recent progress has created many new applications and analysis methods within the last decade. In our review, we present a more encompassing theory; we showcase the new methods that have simplified the complex analysis often associated with MPTR, and we identify and discuss key applications in phase-change, porous, and biological materials, etc. After an introduction to the background, the present theory is shown, and the recently used simplifications are explained. We present our modeling efforts, which show the magnitude of the temperature response as a function of modulation frequency under various conditions. Finally, we present the common experimental setups and the recent analysis techniques and their applications to novel materials.

## 2. Thermodynamic Theory

The signal generated by MPTR is governed by the thermodynamic equations of thermal transport. The thermal response of a sample, for a given laser geometry, is described by Fourier’s law in cylindrical coordinates (*r*, theta, *z*),
(1)$$\frac{k_{r}}{r}\frac{\partial }{\partial r}\left(r\frac{\partial T}{\partial r}\right)+\frac{k_{\theta }}{r^{2}}\frac{\partial^{2}T}{\partial \theta^{2}}+k_{z}\frac{\partial^{2}T}{\partial z^{2}}+q_{v}=C\frac{\partial T}{\partial t}$$
where *T* is the temperature distribution in cylindrical space, $k_{i}$ is the thermal conductivity, $q_{v}$ is the volumetric heat generation, and *C* is the volumetric heat capacity (a product of density and specific heat). The solutions to Equation (1) were first described by Carslaw and Jaeger [29], but since then, other researchers have expanded the solutions. Researchers originally had to limit the analytical solution to a single dimension when they included radiative and convective heat loss [13]; however, later works showed that these losses are negligible when compared to conductive heat transport [30,31]. Current experimental practice often includes a highly absorptive layer. In this case, the volumetric heat generation approaches zero as the heat flux is confined to the surface boundary at z=0. Equation (1) reduces to
(2)$$\frac{k_{r}}{r}\frac{\partial }{\partial r}\left(r\frac{\partial T}{\partial r}\right)+k_{z}\frac{\partial^{2}T}{\partial z^{2}}=C\frac{\partial T}{\partial t}$$

The solutions to the differential equation above are complex and difficult to apply. A typical procedure to reduce cylindrical partial differential equations is to employ Hankel transforms [30]. Applying the zeroth-order Hankel transform to Equation (2) gives
(3)$$\int_{0}^{\infty }r\left(\frac{k_{r}}{r}\frac{\partial }{\partial r}\left(r\frac{\partial T}{\partial r}\right)+k_{z}\frac{\partial^{2}T}{\partial z^{2}}\right)J_{0}\left(xr\right)dr=\int_{0}^{\infty }r\left(C\frac{\partial T}{\partial t}\right)J_{0}\left(xr\right)dr$$
where $J_{0}$ is the zeroth-order Bessel function of the first kind, and *x* is the Hankel transform variable. To avoid confusion, the transformed temperature and heat flux are denoted with a breve to indicate the Hankel-transformed temperature and heat flux. The partial derivative in r is reduced to $x^{2}\overset{˘}{T}$, giving
(4)$$-k_{r}x^{2}\overset{˘}{T}+k_{z}\frac{\partial^{2}\overset{˘}{T}}{\partial z^{2}}=C\frac{\partial \overset{˘}{T}}{\partial t}$$
(5)$$\frac{\partial^{2}\overset{˘}{T}}{\partial z^{2}}=\frac{C}{k_{z}}\frac{\partial \overset{˘}{T}}{\partial t}-\frac{k_{r}}{k_{z}}x^{2}\overset{˘}{T}$$

As MPTR is a frequency measurement, a Fourier transform is applied to Equation (5), which results in the following:(6)$$\frac{\partial^{2}T}{\partial z^{2}}=n\overset{˘}{T};\;n=\sqrt{\frac{k_{r}}{k_{z}}x^{2}+i\frac{\omega C}{k_{z}}}$$
where n is known as the complex wave number and uniquely determines the dimensionality of the heat wave solution. Any anisotropy in the sample is captured by the radial and axial thermal conductivity ratio $k_{r}/k_{z}$. As we are using a modulated heat source, the solution can be expected to be wavelike in nature as well. The general solution of Equation (6) is given by
(7)$$\overset{˘}{T}\left(z\right)=\alpha e^{nz}+\beta e^{-nz}$$
where α and β are constants. The incident heat flux in Hankel space through depth *z* is defined by
(8)$$\overset{˘}{q}\left(z\right)=-\alpha k_{z}ne^{nz}+\beta k_{z}ne^{-nz}$$

The solutions describe temperature and 2*D* heat waves that diffuse through the depth of the material and radially outward. 

The penetration (radially and axially) of this wave is usually defined as the depth where the wave has reduced by 90% and is often approximated by $L_{p}=\sqrt{\frac{2D}{\omega }}$, where *D* is the thermal diffusivity [27]. The penetration depth is visualized in Figure 1. On the left, a laser with modulated intensity is incident on a two-layer sample. The thermal emission is captured using an infrared detector. On the right, the relationship between the modulation frequency and penetration depth is illustrated, showing that a lower frequency results in deeper sample penetration. Equations (7) and (8) can be simplified by relating temperature and heat flux at *z* = *d* to those at *z* = 0 for a single homogenous layer [30,31,32], which results in the following relationship:(9)$$\left[\begin{array}{c} \overset{˘}{T}\left(z=d\right)\\ \overset{˘}{q}\left(z=d\right) \end{array}\right]=\left[\begin{array}{cc} \gamma_{1} & \gamma_{2}\\ \gamma_{3} & \gamma_{4} \end{array}\right]\left[\begin{array}{c} \overset{˘}{T}\left(z=0\right)\\ \overset{˘}{q}\left(z=0\right) \end{array}\right]$$

By substitution of the temperature $\overset{˘}{T}$ and heat flux $\overset{˘}{q}$ at *z =* 0 and *z = d*, the γ parameters are solved as [31]
(10)$$\begin{array}{c} \gamma_{1}=\gamma_{4}=\cosh\left(nd\right),\\ \gamma_{2}=-\frac{1}{k_{z}n}\sinh\left(nd\right),\\ \gamma_{3}=-k_{z}n\sinh\left(nd\right) \end{array}$$

For a multi-layer material with axially varying thermophysical properties, the matrix in Equation (9) is generalized for any number of arbitrary layers [33]. A total transfer matrix $M_{T}$ is made from the product of each individual layer matrix $M_{i}$, as shown below.
(11)$$\begin{array}{c} M_{T}=\left[\begin{array}{cc} A & B\\ C & D \end{array}\right]=M_{N}M_{N-1}\ldots M_{1}\\ M_{i}=\left[\begin{array}{cc} \gamma_{1,i}\left(k_{z,i},n_{i},l_{i}\right) & \gamma_{2,i}\left(k_{z,i},n_{i},\;l_{i}\right)\\ \gamma_{3,i}\left(k_{z,i},n_{i},l_{i}\right) & \gamma_{4,i}\left(k_{z,i},n_{i},l_{i}\right) \end{array}\right]=\left[\begin{array}{cc} \cosh\left(n_{i}l_{i}\right) & -\frac{1}{k_{z,i}n_{i}}\sin h\left(n_{i}l_{i}\right)\\ -k_{z,i}n_{i}\sin h\left(n_{i}l_{i}\right) & \cosh\left(n_{i}l_{i}\right) \end{array}\right] \end{array}$$

The arguments of each matrix $M_{i}$ are dictated by that layer’s thermophysical properties, where $k_{z,i}$, $n_{i}$, and $l_{i}$ are the thermal conductivity, complex wavenumber, and thickness of each layer *i*, respectively. The surface temperature $\overset{˘}{T}\left(z=0\right)$ is solved by substituding $M_{T}$ into Equation (9) and is given by
(12)$$\overset{˘}{T}\left(z=0\right)=-\frac{D}{C}\overset{˘}{q}\left(z=0\right)+\overset{˘}{q}\left(z=d\right)$$
where *C* and *D* are elements of matrix $M_{T}$. For cases where depth *d* is much greater than the penetration of the heat wave, the heat flux approaches zero. Under this condition, the last layer $M_{N}$ is considered semi-infinite and is reduced to
(13)$$M_{N}=\left[\begin{array}{cc} 1 & -\frac{1}{k_{z,N}n_{N}}\\ -k_{z,N}n_{N} & 1 \end{array}\right]$$

The heat flux at the surface $\overset{˘}{q}\left(z=0\right)$ in Equation (12) is determined by the laser’s spatial intensity profile. The spatial intensity of a continuous wave laser beam has a Gaussian distribution, and its propagation is defined by [34]
(14)$$I_{laser}\left(r,z\right)=\frac{2P_{o}}{\pi R_{laser}{\left(z\right)}^{2}}\exp\left(\frac{-2r^{2}}{R_{laser}{\left(z\right)}^{2}}\right)$$
where $R_{laser}\left(z\right)\;$ is the $1/e^{2}$ radius of the electric field of the laser, $P_{o}$ is the laser’s power, r is the radial distance from the beam axis, and z is the axial distance from the beam waist (or minimum beam radius). If the beam is well collimated, $R_{laser}\left(z\right)$ is constant. Assuming no photocarriers are generated, the heat flux through the volume of a material with an absorption coefficient α can be approximated as
(15)$$\begin{array}{c} q_{laser}\left(r,z,t\right)=\frac{2\alpha \left(P_{o}+\Delta P_{o}K\left(t\right)\right)}{\pi R_{laser}^{2}}\exp\left(-\alpha z+\frac{-2r^{2}}{R_{Laser}^{2}}\right)\\ =\frac{\left(F_{laser}+\Delta F_{Laser}K\left(t\right)\right)}{\pi R_{laser}^{2}}\exp\left(-\alpha z+\frac{-2r^{2}}{R_{Laser}^{2}}\right) \end{array}$$
where $F_{Laser}$ and $\Delta F_{Laser}$ are the constant and varying components of the absorbed laser power, respectively. $K\left(t\right)$ modulates the laser power in time at frequency f. The modulation function can be treated as a Fourier series for any given intensity profile [35]. As MPTR is a frequency domain measurement, a Fourier transform is applied to Equation (15). Equation (16) shows the surface heat flux evaluated at a given frequency *ω*.
(16)$$q_{laser}\left(r\right)=\frac{\Delta F_{Laser}}{\pi R_{laser}^{2}}\exp\left(\frac{-2r^{2}}{R_{Laser}^{2}}\right)$$

The Hankel transform is applied to Equation (16) to generate
(17)$${\overset{˘}{q}}_{laser}\left(x\right)=\frac{\Delta F_{laser}}{2\pi }\exp\left(\frac{-x^{2}R_{Laser}^{2}}{8}\right)$$

By substituting Equation (17) into Equation (12), applying the semi-infinite condition, and using the inverse Hankel transform, the temperature response is calculated to be
(18)$$T\left(r,\omega \right)=-\frac{\Delta F_{laser}}{2\pi }\int_{0}^{\infty }x\frac{D\left(x,\omega \right)}{C\left(x,\omega \right)}\exp\left(\frac{-x^{2}R_{Laser}^{2}}{8}\right)J_{o}\left(xr\right)dx$$

The response $T\left(r,\omega \right)$ is a phasor whose amplitude and phase are described by the magnitude $\left|T\right|$ and argument $\tan Im\left(T\right)/Re\left(T\right)$ of the response. Due to the complexity of Equation (18), various simplifications are often used. The following section will explain the commonly applied simplifications. A discussion on the applications of these simplifications is presented in Section 5.

### Theory Simplifications

Equation (18) is reduced further by using the asymptotic form of the Bessel function for small arguments [36] at positions near the center of the beam, resulting in
(19)$$T\left(\omega \right)=-\frac{\Delta F_{laser}}{2\pi }\int_{0}^{\infty }x\frac{D\left(x,\omega \right)}{C\left(x,\omega \right)}\exp\left(\frac{-x^{2}R_{Laser}^{2}}{8}\right)dx$$

Equations (18) and (19) describe the temperature response of the system at the surface. The dimensionality of the heat wave can be accurately considered one-dimensional if the laser spot size is much greater than the radial penetration of the wave [32]. Under this condition, the spherical wave front becomes planar. In essence, the imaginary part of the complex wavenumber n dominates, allowing the approximation $n\approx i\omega C/k_{z}$. The transfer matrix terms *D* and *C* therefore lose dependence on *x* and are treated as constants with respect to *x*.
(20)$$T\left(\omega \right)=-\frac{\Delta F_{laser}}{2\pi }\frac{D\left(\omega \right)}{C\left(\omega \right)}\int_{0}^{\infty }x\exp\left(\frac{-x^{2}R_{Laser}^{2}}{8}\right)dx$$

Equation (20) is an inverse Hankel transform of the laser heat flux. Subsequently, the temperature response for 1*D* heat transfer is given as
(21)$$T\left(\omega ,r\right)=-\frac{\Delta F_{laser}}{\pi R_{laser}^{2}}\frac{D\left(\omega \right)}{C\left(\omega \right)}\exp\left(\frac{-2r^{2}}{R_{Laser}^{2}}\right)$$

Given a small detection window near the beam center or top-hat beam profile, *r* is approximately 0, and Equation (21) can be further reduced to: (22)$$T\left(\omega \right)=-\Delta q_{laser}\frac{D\left(\omega \right)}{C\left(\omega \right)}$$
where $\Delta q_{laser}$ is the AC laser-induced heat flux. For simplicity, a single layer is considered. In this case, the solution takes the following form:(23)$$T\left(\omega \right)=\Delta q_{laser}\coth\left(i\omega C/k_{z}\right)$$

When the penetration depth of the thermal wave is sufficiently in the bulk of the material, the following approximation can be applied:(24)$$T\left(\omega \right)=\Delta q_{laser}\frac{\exp\left(i\frac{\pi }{4}\right)}{e\sqrt{\omega }}$$
where e is known as the effusivity and is given by $e=\sqrt{Ck_{z}}$. For a given fluence and beam radius, the temperature response is inversely proportional to the square root of the frequency. A piecewise function is used to describe the region in Equation (24) when considering multiple layers.
(25)$$T\left(\omega \right)=\left\{\begin{array}{c} \begin{array}{c} \Delta q_{laser}\frac{\exp\left(i\frac{\pi }{4}\right)}{e_{1}\sqrt{\omega }}\;,\;\;\;0<L_{p}<l_{1}\\ \Delta q_{laser}\frac{\exp\left(i\frac{\pi }{4}\right)}{e_{2}\sqrt{\omega }}+\Delta q_{laser}\;R_{1}\;,\;\;\;l_{1}<L_{p}<l_{2}\\ \vdots \end{array}\\ \Delta q_{laser}\frac{\exp\left(i\frac{\pi }{4}\right)}{e_{N}\sqrt{\omega }}+\Delta q_{laser}\;R_{\Sigma }\;,\;\;\;l_{N-1}<L_{p}<l_{N} \end{array}\right.$$
where $e_{i}$ is the effusivity of each respective layer *i*. $R_{\Sigma }$ is the total thermal resistance of the prior layers. For a two-layer sample, $R_{\Sigma }$ is simply the first layer’s thermal resistance $R_{1}$. For samples of three layers and above, the value $R_{\Sigma }$ becomes increasingly complex [37]. While these simplified equations make the analysis simpler, they are reliant on 1*D* transfer; therefore, an effort to understand Equation (18) is needed for cases with high penetration. 

## 3. Modeling

In this section, we showcase two models that explain the theory covered above. First, we present the work that directly implements the equations above in the frequency domain. Second, we present the work that solved the heat equation in the time domain using FEM. 

In our model, we showed the radial temperature response for a three-layer sample, investigated 1*D* validity, and explored how sample anisotropy affected the response. We numerically solved Equation (18) in MATLAB for a three-layer sample. The integration was taken from 0 to $10x_{\max}$, where $x_{\max}=\sqrt{32/R_{laser}}$ [32] in 100 equal steps. The sample was assumed to be isotropic and semi-infinite in both dimensions. The properties of each layer were set to be consistent with known values for Pyromark, chromium, and silicon dioxide and are summarized in Table 1. Pyromark is a highly absorbent paint [38] and was included to maintain the conditions of Equation (2). The sample geometry is shown in Figure 2. 

The spatial dependence of the temperature response is shown in Figure 3a for various modulation frequencies and laser radii. The laser intensity was held constant at $80\;\mathrm{kW}/m^{2}$ to allow direct comparison between cases. As the modulation frequency increased, the radial and axial penetration into the sample increased, with a notable rise in the spatial temperature amplitude. At 1 Hz, we observed 0.4 mm of penetration into the silicon layer. In Figure 3b, the 1*D* (Equation (21)) and 2*D* (Equation (18)) temperature responses at r=0 are shown for various laser radii as a function of the inverse root modulation frequency and penetration depth. As the penetration increased, the heat was radially dampened, causing the response to saturate and thus deviate from the 1*D* response. Subsequently, as the laser radius increased from 5 mm to 20 mm, the radial heat uniformity increased, causing the 2*D* and 1*D* response to be consistent over a larger penetration range. While the sample was assumed to be isotropic, the effect of anisotropy can be studied by modifying the thermal conductivity ratio $k_{ratio}=k_{r}/k_{z}$ introduced in Equation (6). 

Anisotropic materials have uneven radial and axial penetration. When $k_{ratio}<1$, the axial penetration dominates, resulting in reduced saturation. For $k_{ratio}>1$, the radial penetration dominates, increasing the response saturation. Figure 3c illustrates the strength of the anisotropic effect. The $k_{ratio,i}$ of each layer *i* is set to 50 and compared to the isotropic ($k_{ratio}=1)$ case. While the difference is minimal for layers 1 (dashed line) and 3 (dotted), the result for layer 2 (dashed–dotted line) deviates significantly and is caused by the significant thermal conductivity mismatch between chromium and silicon dioxide. In this extreme case, the heat wave is completely absorbed by layer 2, preventing any axial penetration into layer 3; therefore, the anisotropic effects are maximized for thermally mismatched layers. 

Hernandez-Wong et al. [39] computationally presented the transient solution of Equation (24) using FEM analysis, as seen in Figure 4. In Figure 4a, we observe the transient spatial temperature distribution on the aluminum (Figure 4a(i)) and copper (Figure 4a(ii)) disks, respectively, both with a 1 mm thickness and 1 cm radius. The heat flux was modulated using a square profile at a constant frequency of 0.12 Hz, resulting in 1.60 and 1.76 cm of sample penetration, respectively. As the time increased, the temperature increased radially at differing rates due to differences in penetration. The transient temperature at the center can be seen in Figure 4b; it contains a large DC term (largely ignored in the above analysis) and a small AC term whose amplitude and phase can be predicted by Equation (26). The differing thermal properties between aluminum and copper result in different DC and AC behavior. In Figure 4c, the frequency response is shown for both the analytical (full-triangle) and the FEM (empty-triangle) results. The analytical results were formulated using a similar theory to that in Section 2, while the numerical solutions were produced with COMSOL. A discrepancy between these results arises due to the AC component approaching the value of the DC component, resulting in the nonlinear nature of blackbody radiation affecting the accuracy of the analytical model.

## 4. Experimental Setup 

MPTR has three significant elements: (1) a laser-based heat source; (2) radiometric measurement; and (3) signal processing. Figure 5 shows various experimental designs. In the next section, we discuss these setups in detail. 

### 4.1. Heat Source

Continuous-wave (CW) lasers are an ideal source for MPTR, providing a coherent source of light [41]. Commonly used wavelengths range from 1 µm to 450 nm. The CW lasers are intensity modulated over a wide frequency range from 0.5 Hz to 10 kHz. Apart from directly modulating the laser drive current, the beam intensity can be externally modulated [42]. Acousto-optic modulators (AOM) and optomechanical choppers are forms of external modulation which are often used [19,28,43,44,45]. In AOM, the output intensity is proportional to an electric signal voltage through piezo-electric-induced diffraction [46]. While this allows the use of any arbitrary modulation function, AOM can be expensive and can produce higher-order diffractions. An example of its use is seen in Figure 5b. An optomechanical chopper, used in Figure 5c, periodically blocks the beam intensity via a spinning wheel. Although cheap and simple, choppers limit the laser profile to a trapezoidal shape, whose sharpness varies throughout its frequency range [47]. Due to this, many researchers have increasingly opted for direct modulation of the laser drive current [33,37,48,49,50], as seen in Figure 5a. Intensity profiles that are not sinusoidal will generate many harmonics in the temperature response; however, the signal processing of the response reduces the effect of higher-order generated harmonics, as covered in Section 4.3. 

### 4.2. Radiometric Measurement

Radiometry is often accomplished with a photodetector whose output voltage or current is proportional to the incident power and wavelength of the thermal emission. When considering the properties of the thermal emission of a surface at temperature *T*, the upper-bound radiant emittance *j* can be estimated by $j=\sigma T^{4}$ [11], where *σ* is the Stefan–Boltzmann constant. The peak wavelength of the emission can be estimated with Wien’s law [51] by $\lambda_{peak}=\frac{b}{T}$, where *b* is a constant of proportionality equal to 2898 μm K. These two quantities can establish basic photodetector requirements; however, in situations where the radiance is directional, one must use the complete blackbody radiance given by $B_{v}\left(T,\lambda \right)=\frac{2\nu^{2}}{c^{2}}h\nu {\left[\exp\left(h\nu /k_{B}T)-1\right)\right]}^{-1}$ to estimate the incident radiance at the detector. Considering that most materials deviate from the ideal emission, $B_{v}$ is weighted by the material emissivity $\varepsilon \left(\lambda ,\theta ,\Omega \right)$. The properties of the emission collection optics also affect the incident radiance. In Figure 5a,b, parabolic mirrors are used to collect the emission, whose reflectance $R\left(\lambda \right)$ is included (for transmissive optics, such as the lenses, transmittance *T* would be used). A spectral filter can be used to prevent thermal background and laser emission from affecting the generated signal, as shown in Figure 5a,c. The filter transmittance can be included in Equation (26). The detector current is then estimated by including the detection area $A_{d}$ and the detector responsivity $Ɍ\left(\lambda \right)$, which relates the generated current to the incident intensity as a function of wavelength. The detector voltage as a function of temperature is then given by [52]:(26)$$V_{detector}\left(T\right)=G\;A_{d}\int \int \int \varepsilon \left(\lambda ,\theta ,\Omega \right)\;B_{v}\left(T,\lambda \right)\;Ɍ\left(\lambda \right)\;R\left(\lambda \right)\;cos\theta d\lambda d\theta d\Omega$$

The measured thermal signal of the system is then given by Equations (18) and (26). If the emissivity is constant across the wavelengths of interest, Equation (26) can be reduced to
(27)$$V_{detector}\left(T\right)=B\sigma \varepsilon T^{m}+C$$
where *B*, *m*, and *C* are calibration constants that can be found experimentally. T is given by Equation (18) and is composed of a large *DC* component with small *AC* variations; it can be substituted into Equation (27) to show
(28)$$V_{detector}\left(T_{DC}+T_{AC}\right)=B\sigma \varepsilon {\left(T_{DC}+T_{AC}\right)}^{m}+C$$

A Taylor series expansion of Equation (28) is given by [53]: (29)$$V_{detector}\left(T_{DC}+T_{AC}\right)=V_{detector}\left(T_{DC}\right)+\frac{dV_{detector}\left(T_{DC}\right)}{dT_{DC}}\;T_{AC}+\frac{d^{2}V_{detector}\left(T_{DC}\right)}{dT_{DC}{}^{2}\;}\frac{T_{AC}^{2}}{2!}+\ldots$$

The expected *AC* detector signal is then given by:(30)$$V_{AC}\left(T_{DC}+T_{AC}\right)=\frac{dV_{detector}\left(T_{DC}\right)}{dT_{DC}}\;T_{AC}+\frac{d^{2}V_{detector}\left(T_{DC}\right)}{dT_{DC}{}^{2}\;}\frac{T_{AC}^{2}}{2!}+\ldots$$

If $T_{AC}$ is smaller than $T_{DC}$, Equation (31) above is reduced to linear terms: (31)$$V_{AC}\left(T_{DC}+T_{AC}\right)=\frac{dV_{detector}\left(T_{DC}\right)}{dT_{DC}}\;T_{AC}=mB\sigma \varepsilon \;T_{DC}{}^{m-1}\;T_{AC}=ET_{AC}$$
where *E* is the lumped calibration constant and is found experimentally [33]. The signal related to Equation (30) is measured with the signal-processing techniques described in Section 4.3.

Photodetector noise is a significant issue that can arise during the radiometric measurement [54]. When considering the semiconductor materials that constitute conventional infrared photodetectors, any ambient light can excite electrons above their small bandgap and generate photocarriers that result in additional photodetector current, which is often called dark current [55]. Detectivity $D^{*}$ is often used as a method to normalize the SNR of the detector and is given by
(32)$$D^{*}=\frac{{Ɍ}_{p}\sqrt{A_{d}\Delta f}}{i_{d}}$$
where ${Ɍ}_{p}$ is the peak detectivity; Δf is the device bandwidth; and $i_{d}$ is the dark current, which is a function of temperature.

Mercury–cadmium–telluride (HgCdTe) photodetectors are often used for their wide responsivity (2–10 µm) and comparably low integration time; however, they suffer reduced detectivity at room temperature from large dark currents [56]. Liquid nitrogen is often used to significantly reduce thermally induced currents and boost HgCdTe detectivity. Beyond noise, infrared detectors also suffer from other obstacles. When considering MPTR capability, Battaglia [49] identified photodetector response time as the main limiting factor. As a solution to this issue, photodetectors using electron confinement in quantum wells [57], nanowires [58], and quantum dots [59] have been proposed, showing carrier lifetimes in the ps range. Although these designs are promising, they often suffer from reduced SNR due to narrow absorption bands and manufacturing difficulties [56]. Chen et al. [60] identified potential 2*D* material-based detectors and cited examples with low noise (33 fW Hz^−1/2^) and 1 GHz operation; however, 2*D* materials have low intrinsic absorption, resulting in reduced responsivity to blackbody sources; hence, they are inappropriate for use in MPTR. 

### 4.3. Signal Processing

Typically, the SNR of the recorded thermal signal is small, making direct analysis difficult; thus, signal processing techniques are needed to extract the required information for analysis [45]. Common among most MPTR implementations [28,45] is the use of lock-in amplifiers for signal processing. Lock-in amplification is a homodyne detection technique that modulates an input signal at frequency $f_{s}$ with a reference signal at frequency $f_{ref}$. In the frequency domain, this modulation results in an output with components centered at $\left(f_{s}+f_{ref}\right)$ and $\left(f_{s}-f_{ref}\right)$. The reference signal frequency is controlled to match the input signal, thus reducing the output to DC and $2f_{s}$ components [35]. The magnitude of these components is proportional to the amplitude and phase difference of the input and reference. A lowpass filter is used to isolate the DC component, resulting in a strong, noise-resistant signal that can be described by $V_{lock}=\frac{1}{2}V_{AC}\cos\left(\theta_{s}-\theta_{ref}\right)$ [61]. By repeating the detection with a 90° phase-shifted reference, the phase dependence can be eliminated, allowing separate amplitude and phase measurements. While this signal processing technique has stood the test of time, it comes with its challenges. Under correct operation, the cut-off frequency of the lowpass filter must be kept sufficiently below the 2*f_s_* harmonic mentioned above. For low frequency (<1 Hz) measurements, the requirement can significantly extend the total experiment time [62]. While the lock-in processing can be implemented with software, it is often applied with a Stanford Systems SR830 Lock-In Amplifier. Filtering can be employed before the lock-in to further reduce noise, but its effectiveness is unclear [63].

## 5. Analysis

As mentioned in Section 2, the complex temperature response T is described by its magnitude and phase, which are captured in the photodetector voltage described by Equation (30). The lock-in techniques described in Section 4.3 directly measure the magnitude and phase of this signal. Analyzing this signal and the expected response allows the thermophysical properties to be measured. Phase analysis requires minimal system calibration, while magnitude analysis requires the estimation of the calibration of constant *E* in Equation (31). The previous techniques focused on phase measurements to avoid the calibration; however, new work has changed this focus. Initial efforts to avoid magnitude calibration employed the parametric sweeping of variables such as thickness [64]. A recent technique used nonlinear effects and heterodyning to overcome the calibration issues often found in MPTR. While heterodyning has been used in lock-in thermography [65], Chirtoc [53] extended the concept to MPTR. In this method, two lasers (of different wavelengths to avoid interference) are superimposed at the sample surface; this can be used to cancel out calibration constants in the MPTR signal. While this approach is interesting, it is difficult to apply, requiring two laser sources (each with its own modulation source) and lock-in amplifiers. Zeng et al. [33] provided two solutions: the omission of the calibration constants by ratio measurement to a known reference (often used in pyrometry [66]), and direct calibration using a separate pyrometer. Figure 6a shows their calibrated temperature response for a two-layer sample consisting of coated 304 stainless steel disks (4.25 mm radius). Three different solar-absorbing coatings were studied. The coating thickness ranged from 11 to 45 µm, while the steel substrate was 1 mm thick. We see two distinct linear regions form; these are associated with the coating and bulk as the modulation frequency decreases. By using the linearized relationships shown in Figure 6c, the slopes of these regions were used to produce coating thermal conductivity measurements across a wide temperature range, as shown in Figure 6b. While the solutions proposed by Zeng et al. are attractive, they have their drawbacks. Their ratio measurement depends on the validity of 1D heat transfer [33], thus limiting the experimental scope and applicability by requiring the laser radius to be much larger than the radial penetration. Pyrometer-based calibration suffers from inaccuracies due to limited bandwidth from low response times (>1 ms) and requires well-known sample emissivity. 

Hua et al. [67] avoided calibration by fitting to a spatial MPTR signal. The spatial signal was produced by scanning the laser across the sample surface; however, the measurement required the consideration of diffraction and nonlinear effects, complicating the analysis. Pawlawk et al. expanded Nordal’s work by providing a simplified methodology to measure absorption [68]. Fuente also produced simplified methods to obtain the absorption and thermal diffusivity of semi-transparent samples of small thicknesses and showed sufficient agreement with modulated pyroelectric (PPE) techniques [69]. New approaches to analyzing MPTR data have streamlined the measurement and produced increasingly accurate results; thus, the combined work of the authors cited has led to the application of MPTR to a wider range of novel materials. An overview of these applications is provided below. 

## 6. Applications

Within the last decade, MPTR applications have grown vastly. The noninvasive and nondestructive nature of the measurement makes it appealing to various fields, from manufacturing to biomedicine. In the next section, we focus on new applications of MPTR that have evolved since the last review by Pham et al.

Phase change materials (PCMs) have been a growing area of research, with applications in mechanical and electrical engineering. As the name suggests, these materials have temperature-dependent, reversible structures that switch between chaotic amorphous and crystalline forms [64,70]. Initial studies at the Laboratoire National de Métrologie et d’Essais (LNE) used thermal resistance-based measurements for PCM characterization [40]. The thermal resistance and film thickness relationship is shown in Figure 7a and is similar to Equation (25). Figure 7b shows the same measurements for the ambient temperatures of $71{\;}^{o}C-370{\;}^{o}C$ and the associated physical structure of the PCM. This relationship was used to measure effusivity for different GeTe film thicknesses. Battaglia et al. measured the thermal boundary resistance across the material’s transition temperature, comparing MPTR, pulsed photothermal radiometry (PPTR), and time-domain thermoreflectance (TDTR). They found that MPTR could penetrate the deepest into their samples, but PPTR was a better alternative to TDTR [49]. Other researchers focused on refining the work at LNE by establishing the phonon density of GeTe films on silicon substrates, separating the lattice and electron thermal conductivity contributions [71]. Through these thermal resistance studies, MPTR has been identified as a possible way to measure thermal boundary resistance, which is also known as Kapitza resistance. Thermal boundaries exist across any interface of different materials. At the macroscale, this boundary can result from reduced thermal contact, but more fundamentally, it arises from the scattering of phonons at the interface [72]. The Kapitza resistance of titanium on silicon and stainless-steel substrates was measured by Horny et al. [73] using MPTR. In their study, they determined that increased Kapitza resistance could be attributed to Debye temperature mismatch at the material boundary.

In the studies above, the ambient temperature was controlled by placing the sample in an enclosed furnace, which enabled high-temperature measurements, as seen in Figure 5b; however, the remote sensing capabilities of MPTR can enable other heated sample methods while avoiding damage to sensitive equipment. In the investigation of high-temperature solar-absorbing coatings, Zeng et al. showed an internally heated sample that could be used across a range of 100–600 °C to extract the thermal properties of the coating and bulk [33], as seen in Figure 6b and mentioned in Section 5. However, their application only featured coatings and bulk materials of similar thermal properties, reducing the ability to extrapolate results to other materials. Other coatings have also been studied. Sol-gel-based coatings have various corrosion inhibition applications; however, these composites of organic and inorganic polymers often suffer from unwanted porosity due to cracks and other setting defects [74]. Chrobak et al. [48] used both beam deflection spectroscopy (BDS) and MPTR to measure the thermal properties of sol-gel for different additions of cerium/zirconium and thus determined a measure of porosity. They found reduced thermal conductivity as the coating porosity increased; moreover, they found that BDS and MPTR produced measurements that were in close agreement. MPTR’s sensitivity to porosity has also been used to characterize defects in other materials.

MPTR has recently been applied to characterize laminated material quality. Laminated materials can contain defects, either through manufacturing errors or damage, which can result in delamination of the layers. Salazar [75] showed that the depth and size of these delaminations could be measured with MPTR. Figure 8a shows the embedded delamination geometry. The findings are summarized in Figure 8b, which details how the temperature magnitude and phase are affected as a result of delamination width. The associated thermal resistance was found by analysis of the magnitude and phase, which allowed generalized delamination depth and width measurements. 

We have also contributed to the study of porous materials by measuring thermal conductivity of particle beds. In our experiment, we used binary-size particle mixes to vary total particle bed porosity. The resulting thermal conductivity changes were measured with MPTR; however, significant thermal resistance at the wall–particle interface presented challenges. Probing past the interface while maintaining 1D heat transfer required increasingly unrealistic beam radii, and thus, a 2*D* approach was used. The results of this work have possible applications in concentrating solar power (CSP) plant control [76]. Other CSP applications have been considered. CSP plants also often feature fluidized thermal management and transport systems to move thermal energy from sources to storage and vice versa. Zeng et al. [37] described a method of in situ monitoring of the thermal properties associated with a fluid using MPTR, citing CSP as a great application. MPTR has also gained traction in other fields that benefit from noninvasive monitoring.

Biomedical engineering applications have also increased in popularity. Ablation is an emerging technique used to target and destroy problematic tissue; however, issues with the precision of rapid heating create a safety hazard [77]. Kosik et al. [78] presented a solution using MPTR to monitor the ablative process and mitigate safety concerns. MPTR has been used in other biological monitoring applications as well. Cheng et al. [79] demonstrated high-resolution chemical imaging of tissue. While the discussion of the biological applications presented by Kosik et al. and Cheng et al. was minimal, future publications on the subject are expected to provide further detail.

The flexibility afforded by MPTR allows its use in various fields and applications. The noninvasive technique is not limited to macroscopic thermal characterization but can be used to study many other phenomena. Further work will continue to develop this technique, extending its use cases and increasing accuracy and reliability, resulting in the emergence of extensive applications.

## 7. Conclusions

MPTR has been shown to make effective thermal measurements with a simple experimental design. It has grown significantly from its first proposed applications and has been applied to a wide variety of materials for remote sensing, measurement, and characterization. The thermodynamic theory and analysis technique associated with the technique has developed significantly from previous works, where only the phase of the thermal response was used to study the material properties, to include amplitude response. The greater understanding of radial penetration against depth penetration has allowed clever engineering that simplifies the analysis time. Many different experimental setups have been demonstrated and have been shown to produce accurate results. The accuracy of these results has also been validated using a variety of other techniques, such as time-domain thermoreflectance [49] and deflection spectroscopy [48]. While these differing experimental designs produce similar results, some argue that the non-standardized apparatus design makes any comparisons inconclusive [45]. Other concerns, such as the lack of standard samples and uncertainty in the analysis, have started to be addressed [33,37,48,66,75]. Measurable sample parameters are mainly determined by modulation frequency bandwidth and resolution [66,75]; therefore, infrared photodetectors have become the main limiting factor as their low response time limits the usable upper-frequency range of MPTR [49]. While photodetectors with significantly reduced response times have been demonstrated [56,60], they often suffer from reduced detectivity, limited absorption, and/or manufacturing difficulty. Other workarounds are likely to be employed soon as interest in MPTR continues to grow. As MPTR has matured, its application to materials has grown to include various PCM materials and coatings. The demonstrated sensitivity to porosity has great potential, but further understanding is needed to relate MPTR to the various modes of heat transport present in porous systems. Future work will likely show greater applications in biomedical engineering, producing simpler and safer methods of targeted tissue necrosis and imaging. 

## Figures and Tables

**Figure 1 sensors-23-04935-f001:**
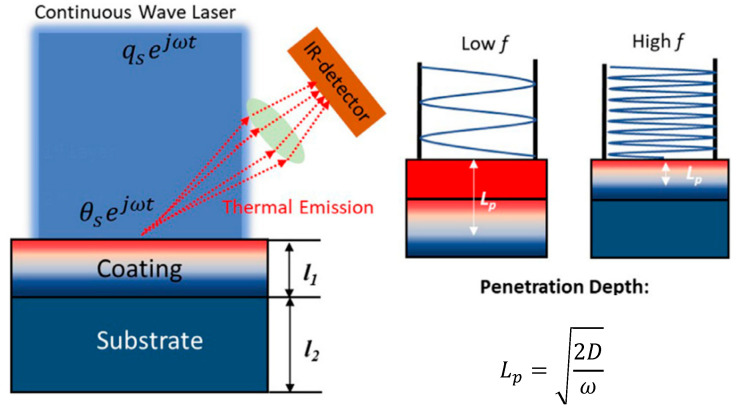
Heat wave penetration in a 2-layer sample [33]. Reproduced with permission.

**Figure 2 sensors-23-04935-f002:**
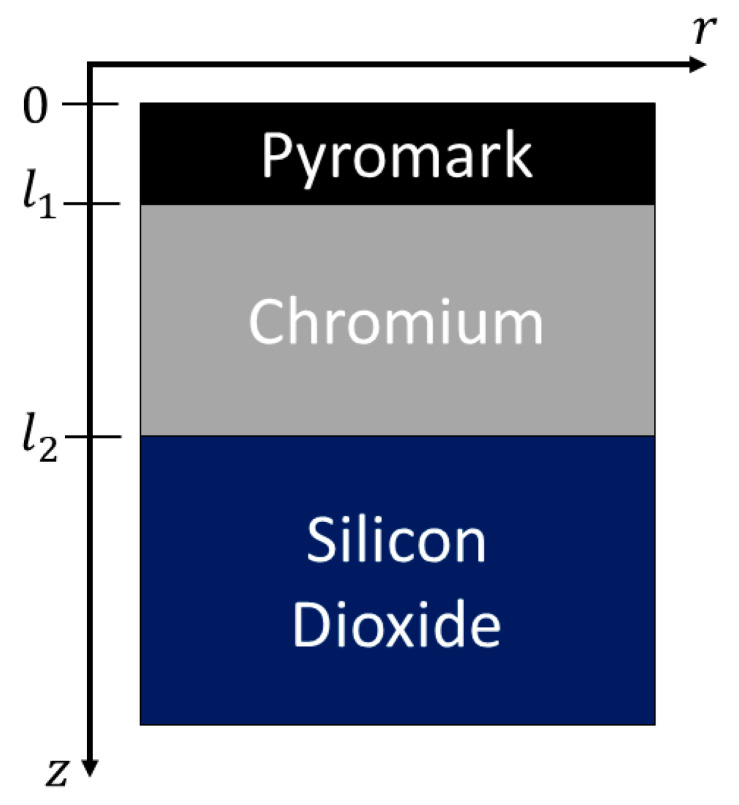
Schematic of the 3-layer model showing layer dimensions for Pyromark, chromium, and silicon dioxide.

**Figure 3 sensors-23-04935-f003:**
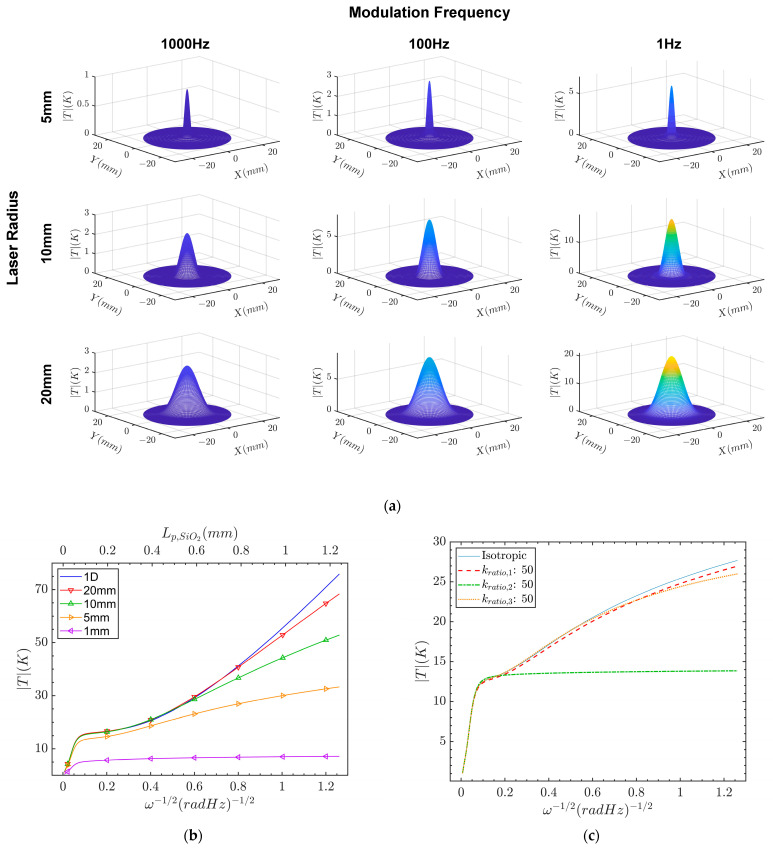
Plots of the 2*D* model under the following conditions: (**a**) spatial temperature response for select frequency and laser radius values; (**b**) comparison of 1*D* and 2*D* models with varying laser radius sizes against the inverse square frequency and penetration depth; (**c**) 2*D* model for 4 mm beam radius with varying $k_{r}/k_{z}$ ratio for each layer.

**Figure 4 sensors-23-04935-f004:**
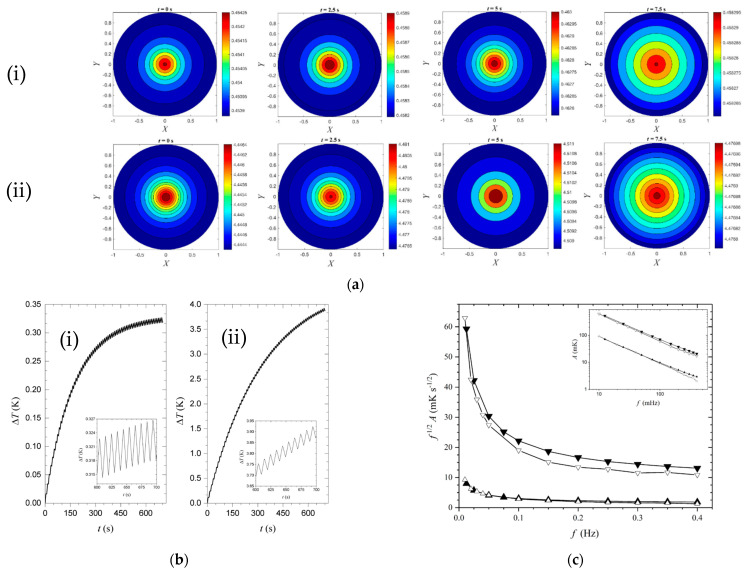
Transient simulation results from [39] show: (**a**) spatial temperature distribution at the sample surface for aluminum (**i**) and copper (**ii**) samples; (**b**) transient surface temperature for aluminum (**i**) and copper (**ii**) samples, respectively; (**c**) amplitudes of the temperature variation against frequency, comparing analytical (full triangle) and numerical FEM (empty triangle) solutions for aluminum (upwards-facing triangle) and copper (downwards-facing triangle). Reproduced under CC BY-NC-ND 4.0.

**Figure 5 sensors-23-04935-f005:**
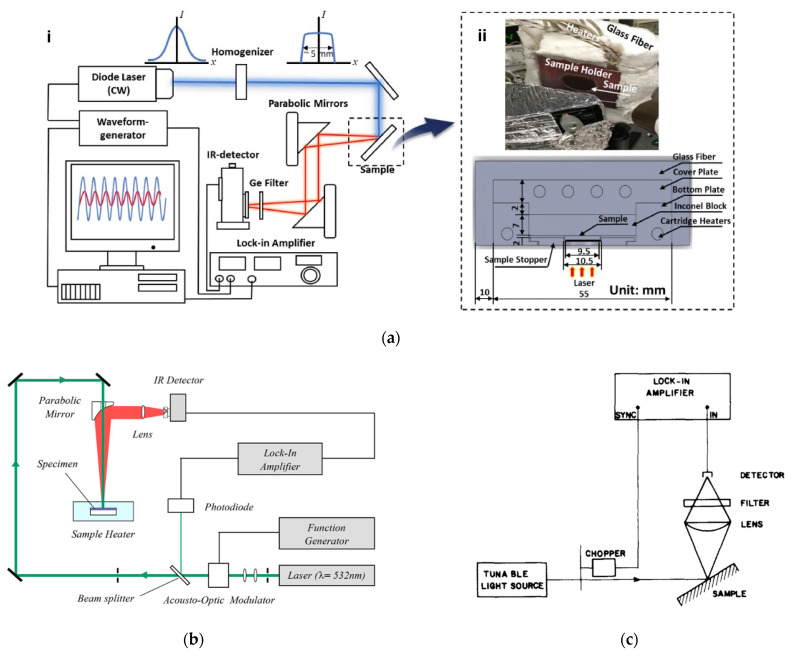
Various examples of MPTR experimental setups: (**a**) setup using direct sinusoidal modulation of a 445 nm CW diode laser (**i**) and internally heated sample holder (**ii**) [33]; (**b**) AOM externally modulated 532 nm CW laser producing a square profile with an externally heated sample [40]; and (**c**) optomechanical chopper modulation with lens optics [13]. Reproduced with permission.

**Figure 6 sensors-23-04935-f006:**
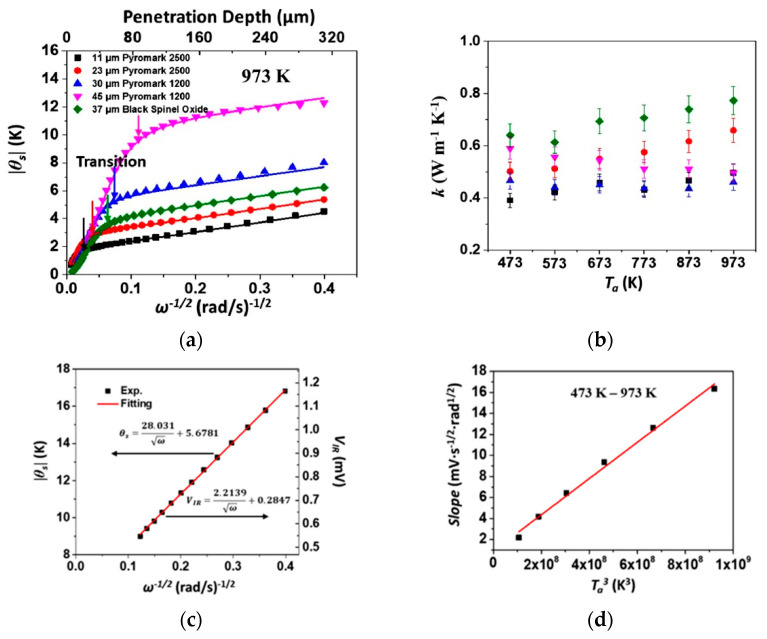
MPTR measurements on 2-layer samples [33]: (**a**) temperature amplitude response showing distinct linear regions for coatings on SS304; (**b**) extracted thermal conductivity for coatings; (**c**) linearized response in the bulk region with slope equivalent to effusivity; (**d**) linear region slope change according to sample temperature on Pyroceran 9606. Reproduced with permission.

**Figure 7 sensors-23-04935-f007:**
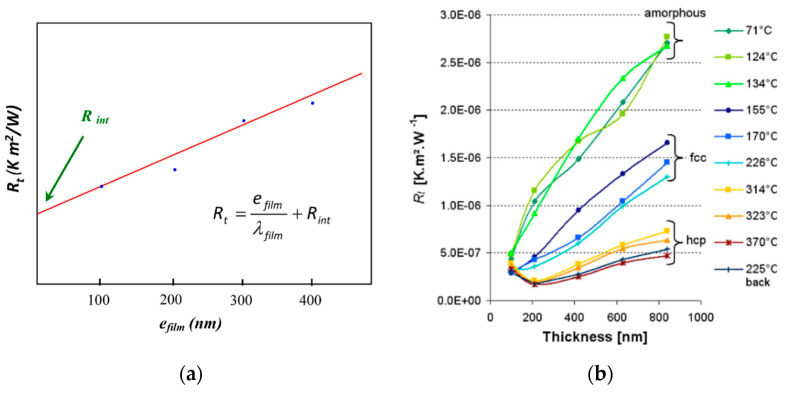
(**a**) Thermal resistance of film compared to film thickness; (**b**) thermal resistance versus film thickness at various temperatures showing phase dependence on thermal properties of GeTe films [40]. Reproduced with permission.

**Figure 8 sensors-23-04935-f008:**
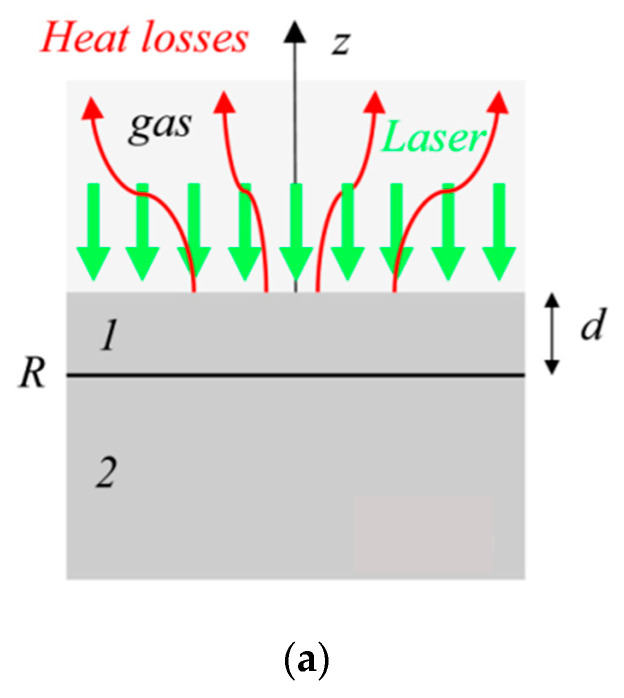
(**a**) Graphic showing embedded thermal resistance from delamination where 1 represents the area before and 2 the area after the delamination; (**b**) temperature amplitude (**left**) and phase (**right**) measurements showing distinct signals for various delamination widths where the arrow highlights the start of the delamination [75]. Reproduced with permission.

**Table 1 sensors-23-04935-t001:** Values used for the 3-layer model.

Property	Layer 1 Pyromark ^1^	Layer 2 Chromium	Layer 3 Silicon Dioxide
Density $\rho \;\left(\mathrm{kg}/m^{3}\right)$	-	7190	2270
Specific Heat $c_{p}\;\left(J/\mathrm{Kkg}\right)$	-	459.8	1000
Heat Capacity $C\;\left(J/{\mathrm{Km}}^{3}\right)$	$1.23\times 10^{6}$	$3.31\times 10^{6}$	$2.27\times 10^{6}$
Thermal Conductivity $k\;\left(W/\mathrm{mK}\right)$	0.5	67	1.1
Thickness $l\;\;\left(\mu m\right)$	50	600	−

^1^ Pyromark measurements from [33].

## Data Availability

No new data were created or analyzed in this study. Data sharing is not applicable to this article.

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
