# Peer review of "Recent Progress in Modulated Photothermal Radiometry"

_sensors, 2023, doi:10.3390/s23104935_

Round 1

Reviewer 1 Report

Well written and clear!

A few typos are present, for example in the Introduction on line 8 there is a repeated use of the word "the" and a fullstop that should't be there!

Author Response

Thanks you for your comments. We will go through the document more carefully and fix the typos.

Reviewer 2 Report

The manuscript titled “Recent Progress in Modulated Photothermal Radiometry” authored by Javier Corona and Nirmala Kandadai reviews the progress in the field of non-invasive technique, Modulated Photothermal Radiometry. The topic is an interesting one and the authors have tried to give an overview. However, the manuscript has many flaws in it. Some of these are mentioned below:

·         First and foremost concern is the language. There are serious errors in the manuscript. For example: On page 1, line 18-19, the sentence is incorrect. In line 24, there are two “fullstops” used in the middle of the sentence. In line 26, “then” is used in place of “than”. And the list of such mistakes goes on and on. It seems there has not been much effort put in the proof-reading of the manuscript before submission.

·         The abstract needs major re-writing. Apart from the incomplete sentences, the abstract is not reflecting the purpose of the review well.

·         On page 1, line 29, the authors are talking about the 3ω technique without clearly mentioning what exactly is it. Someone new to the field would definitely like to know about it first.

·         Although the entire review is on Modulated Photothermal Radiometry, the authors have suddenly used the abbreviation “MPTR” on page 2, line 57. Again, on the same page, line 86, they are writing “Modulated photothermal radiometry (MPTR)”. My suggestion would be the club sections 1 and 2. The background may be a part of the introduction itself.

·         Again, the sections have not been numbered carefully. The “Thermodynamic Theory” has been numbered as Section 3 on page 3 while on page 9 the “Experimental setup” has also been numbered as section 3. Further, on page 11 and 12, the subsections have been incorrectly numbered as section 2.1, 2.2, etc.

·         In the “Thermodynamic Theory” part, the authors have covered the thermodynamic part quite well but this information is well known and is widely available in the existing literature.

·         Table 1 depicts the values used for a 3-layer model in Figure 2. There should be a schematic representation of the 3-layer model for better clarity of understanding.

·         The “Conclusion” according to me is inconclusive and needs re-writing.

On the basis of the aforementioned points, the manuscript cannot be accepted for publication. Hence, I recommend the manuscript to be rejected.

Reviewer 3 Report

In the attached PDF file

Reviewer 4 Report

The abstract of the manuscript is full of incomplete, incoherent or very short sentences like  “A brief history of the technique is covered and its proposed applications” (line 10/11), “Experimental apparatus design is discussed” (line 12). There are also several incomplete sentences, see for example “Thermal processes are fundamental to electrical, mechanical, chemical, biological, fields” (line 17). There are so many redundant and meaningless sentences like “The temperature distribution, rate of change of temperature with respect to time, and diffusion are crucial to effective performance of materials perform” (line 18). I am really astonished by sentences like this: “Researchers [3] Shanks et al. positioned a heater on one end of a silicon sample and placed thermocouples along its depth. and measured the. the thermal conductivity and diffusivity directly” (22/23). These and many other problems make the manuscript very difficult to read. These carelessly written sentences imply that either the manuscript is written by AI or it was not given due attention. I suggest the authors to reread their work before sending out.

I tried to push reading the manuscript disregarding these hindrances. However, I cannot follow any story in a given paragraph. It seems to me that the introduction is just coped and pasted from several works so that it has no flow of any idea. Within a given paragraph, the authors jump from here to there without sticking to any single story. This makes this work unfinished one and even not has been done well at all.

Because of the above facts, I stopped reading the manuscript for the sake of my health and time. I can’t find anything interesting in this manuscript and I suggest that the authors need to resubmit after rereading, rewriting and rearranging their poorly written article. At this stage, this manuscript should be rejected, with a possibility of resubmission after major revision.

Reviewer 5 Report

Corona at al. reported “Recent Progress in Modulated Photothermal Radiometry”. A brief history of the technique is covered and its proposed applications. Models are discussed detailing its evolution and relevant approximations used. Experimental apparatus design is discussed. Recent analysis techniques and applications are described. ThereforeI think this manuscript can be published after noticing the following issues:

1.     Make sure all abbreviations are written out in full the first time used. This is particularly important in the abstract and in the conclusions, but work through the entire ms carefully from this perspective.

2.     Better quality images are recommended, especially Figure 3.

3.     References are added to the summary section. In the discussion section, we have analyzed whether it is necessary to increase.

4.     In most review, the summary and outlook section are important section. Thus, the authors should discuss further the summary and outlook of this minireview.

5.     This review paper appears to lack several tables that provide systematic overviews on various aspects covered in this review paper. These tables are of great benefits to the readership for summarization, benchmarking and comparisons.

6.     In the “Introduction” section, the authors need to justify why the review article fills a critical gap in the field, is indeed in need and timely. Are there any existing review articles on the topic?

7.     As a review, more relevant references about photothermal reaction need to be added such as J. Mater. Chem. A, 2018, 6, 24740–24747; Applied Catalysis B: Environmental 218 (2017) 700–708; Applied Catalysis B: Environmental 181 (2016) 779–787.

Round 2

Reviewer 2 Report

The revisions made in the manuscript are fine. The manuscript may now be accepted for publication.

Author Response

We thank you and appreciate your comments and effort. We believe our review has improved consequently.

Reviewer 4 Report

The authors revised the manuscript substantially and thus I recommend publication. 

Author Response

(The authors gave the same response as above.)
